# Outer Membrane Vesicles from *Acinetobacter baumannii*: Biogenesis, Functions, and Vaccine Application

**DOI:** 10.3390/vaccines12010049

**Published:** 2023-12-31

**Authors:** Zheqi Weng, Ning Yang, Shujun Shi, Zining Xu, Zixu Chen, Chen Liang, Xiuwei Zhang, Xingran Du

**Affiliations:** 1The Second Clinical Medical School, Nanjing Medical University, Nanjing 210011, China; zheqiweng@163.com (Z.W.); ssj0212@126.com (S.S.); ziningxxu@163.com (Z.X.); czx909821823@163.com (Z.C.); lianglcnjmu@163.com (C.L.); 2Department of Respiratory and Critical Care Medicine, The Second Affiliated Hospital of Nanjing Medical University, Nanjing 210011, China; yangning689@163.com; 3Department of Respiratory and Critical Care Medicine, The Affiliated Jiangning Hospital with Nanjing Medical University, Nanjing 211100, China

**Keywords:** *Acinetobacter baumannii*, outer membrane vesicles, biogenesis, antibiotic resistance

## Abstract

This review focuses on *Acinetobacter baumannii*, a Gram-negative bacterium that causes various infections and whose multidrug resistance has become a significant challenge in clinical practices. There are multiple bacterial mechanisms in *A. baumannii* that participate in bacterial colonization and immune responses. It is believed that outer membrane vesicles (OMVs) budding from the bacteria play a significant role in mediating bacterial survival and the subsequent attack against the host. Most OMVs originate from the bacterial membranes and molecules are enveloped in them. Elements similar to the pathogen endow OMVs with robust virulence, which provides a new direction for exploring the pathogenicity of *A. baumannii* and its therapeutic pathways. Although extensive research has been carried out on the feasibility of OMV-based vaccines against pathogens, no study has yet summarized the bioactive elements, biological activity, and vaccine applicability of *A. baumannii* OMVs. This review summarizes the components, biogenesis, and function of OMVs that contribute to their potential as vaccine candidates and the preparation methods and future directions for their development.

## 1. Introduction

*Acinetobacter baumannii* is an iatrogenic pathogen. It can cause various infections, including pneumonia, meningitis, sepsis, urinary tract infections, and wound infections [1]. *A. baumannii* infection seriously threatens patients’ lives, especially those in the intensive care setting. Based on its treatability, mortality, health-care burden, the increase in drug resistance, and other criteria used to assess its bacterial harm, carbapenem-resistant *A. baumannii*, with a score of 91.0%, has been categorized as one of the top three critical and notorious pathogens in the list published by the World Health Organization (WHO) [2]. Antibiotic resistance and health-care burden are particularly prominent in the scoring of the criteria. The WHO priority list highlights areas where urgent interventions are needed at a global level. It has been estimated that *A. baumannii* is responsible for nearly 12% of all hospital-acquired infections worldwide [3]. Due to the lack of effective medical therapies, this situation is on the verge of significant deterioration.

Multiple interventional and prevention strategies have been implemented in clinical settings. Effective hand hygiene compliance is the general strategy for reducing *A. baumannii* infection in health-care settings. In addition, active surveillance cultures, patient isolation, and monitoring are effective control measures [4]. However, *A. baumannii* shows powerful resistance to antibiotics and high mortality rates. It is reported that cases caused by multidrug-resistant isolates attain a mortality rate as high as 44% [5]. Studies have shown that the mortality rate from *A. baumannii* ventilator-associated pneumonia (VAP) and bloodstream infection can reach levels of up to 70% and 43%, respectively [6]. Therefore, the practices mentioned above to prevent and control *A. baumannii* infection are limited in their effect, especially for hospitalized and vulnerable patients. For *A. baumannii*, antibiotic resistance is a severe and cross-sectional problem affecting human beings, the environment, and society. The need for measures focusing on improving infection prevention is compelling. Targeted prevention and control measures for *A. baumannii* are urgently needed to reduce the epidemics and mortality rate of *A. baumannii* in hospital settings. Vaccines are one of the most robust preventative routes for bacterial infections; therefore, vaccines against *A. baumannii* are considered as an alternative to traditional therapeutic drugs because they can significantly reduce the interference of drug resistance.

A subunit vaccine is a type of vaccine that involves active fragments of the pathogen to stimulate a protective immune response. The composition of subunit vaccines is relatively simple and clear, consisting of proteins, polysaccharides, or peptides. Subunit vaccines can rationally engineer and optimize the epitope structure, thus showing chemical stability [7]. Compared with whole-cell vaccines, subunit vaccines have the advantages of purity, stability, and efficiency. Many researchers have explored potential antigens for the subunit vaccine against *A. baumannii*. Among them, outer membrane proteins [8], fimbrial proteins, and capsular polysaccharides are the most popular candidates. These candidates are integral components of *A. baumannii* and can effectively induce the immune response. Outer membrane vesicles are also promising vaccine design candidates. The OMVs, secreted by Gram-negative bacteria, comprise many bacterial components, including phospholipids, proteins, lipopolysaccharides (LPS), and nucleic acids [9,10]. So far, OMV vaccines against meningococcal disease and gonorrhea have been employed clinically [11,12]. A series of research studies have confirmed that the OMV vaccines against *A. baumannii* can increase the survival rate in animal models [13,14].

To the present time, there has been no comprehensive generalization of *A*. *baumannii* OMVs in subunit vaccine applications. In this review, we summarized the preparation method, pathogenic mechanism, immunogenicity, and immune protective effect of OMVs from *A. baumannii*. On this basis, we realized the potential application of OMVs in *A. baumannii* vaccines while also pointing out their shortcomings. In addition to the application of *A. baumannii* vaccines, the auxiliary role of OMVs in other vaccines is also worth studying. This review provides more comprehensive research directions for OMV vaccine development, including preparation techniques and the efficacy enhancement of other vaccines. More profound research into the immune mechanism can provide a theoretical basis for developing the OMV vaccine and providing vital protection for patients, effectively improving their survival rate and quality of life.

## 2. Preparation of *A. baumannii* OMVs

Gram-negative bacteria release the outer membrane and periplasm through the production of OMVs. The formation of the OMVs results from multiple factors, including the lipoprotein (LPP) connection, the curvature change in the membrane, and the electric charge of LPS (Figure 1). Nevertheless, its exact assembly mechanisms and cargo selection processes have yet to be verified [15].

The cell wall of *A. baumannii* is mainly composed of the outer membrane (OM) and the inner peptidoglycan (PG). The outer membrane lipoprotein (LPP) covalently connects the OM and PG layers via LPP-PG. The relationship between LPP and OMV biogenesis was first discovered in 1976. Hoekstra et al. determined the expression level of both LPPs in the outer membrane of *E*. *coli* and those involved in the vesicles. It was reported that fewer LPPs were contained in the vesicles than in the outer membrane [16]. Based on this discovery, the study found that the crosslinking of two layers weakens when the covalent bond is defective or the local LPP is hydrolyzed. The outer membrane grows faster than the peptidoglycan layer, protruding outward to form an OMV.

The outer membrane (OM) of *A. baumannii* is a bilayer. Based on this biological property, studies have shown that the OM will bend and deform with an increase in the curvature of the bacterial OM [17]. The deformed membrane will protrude outwards to form spherical vesicles. The aggregation of phospholipids can cause changes in the curvature. Research has confirmed that the silence or deletion of the VacJ/Yrb genes leads to an increase in the accumulation of phospholipids, leading to asymmetric dilation of the OM [18]. Additionally, during the synthesis of the peptidoglycan layer, its high concentration causes protrusions in the OM and indicates the occurrence of OMVs.

LPS is also an essential factor in the formation of the OMVs. *Pseudomonas aeruginosa* produces two types of LPS: negatively charged LPS and neutrally charged LPS. Under oxidative stress conditions, OMVs released by *Pseudomonas aeruginosa* are mainly comprised of negatively charged LPS. Therefore, causing the repulsion of negatively charged LPS in the outer membrane was proposed to help repel or reduce the formation of OMVs. In addition to the electric charge of LPS, its modification also contributes to OMV formation, including the mutation of genes that are responsible for core polysaccharide synthesis [19].

Other components in the OMVs have been demonstrated to play an essential role in their biogenesis. OmpA interacts with peptidoglycan and the outer membrane to determine the sites where vesicles can form. It is possible that more than one molecule, such as LPS, outer membrane proteins, and phospholipids, could all contribute to vesicle formation [19].

OMVs play a crucial role in the survival of bacterial cells under abnormal and extreme environmental conditions [20]. Therefore, changes in the surroundings will have an apparent effect on the yield of OMVs. For example, high temperatures can enhance the fluidity of the membrane, speeding up the production of OMVs [21,22]. The pH fluctuation also regulates membrane integrity and can indirectly affect OMV production via modification of LPS [18]. For example, PmrAB is a system existing in Citrobacter rodentium and is regulated by low pH. Once PmrA is activated, it stimulates the differential expression of *pmrC* and *cptA*, both of which catalyze LPS modifications [23]. Based on its susceptibility to the surroundings, the expression of OMVs can be modified artificially to increase the expression level of OMVs. It is a critical part of the manufacture of OMV vaccines.

Vesicles differ in varying circumstances, such as production by different bacteria or exposure to different environmental conditions. In *E. coli*, we can see the selective packaging of LPP into OMVs, which is guided by lipoprotein export signal amino acid sequences. In *P. gingivalis*, a conserved C-terminal domain (CTD-family) can attach to the outer membrane through a connection with negatively charged LPS. As mentioned before, the accumulation of negatively charged LPS is one of the mechanisms of OMV formation. Therefore, it is not surprising that CTD-family proteins are enriched in OMVs. No conserved protein was packaged in OMVs from different species, and proteins play a vital role in the function of OMVs. Therefore, proteins that are contained in different OMVs depend on their function [24].

Overall, the formation of OMVs seems to depend on diverse mechanisms and differs between species. The procedures for OMV biogenesis and cargo are overlapping [25]. OMVs produced by various routes are expected to bear different loads. Until now, the mechanism of how OMVs package their respective cytosolic molecules, such as DNA, remains unclear. Discerning the exact mechanism of the OMV formation and cargo can help us comprehend their biological functions, improve production efficiency, and increase the specificity of the package in OMV.

### 2.1. Components

OMVs are spherical nanovesicles with an average diameter of 20–200 nm, secreted from the bacterial OM through budding [17]. Therefore, their composition is theoretically similar to the bacterial OM. The composition of OMVs includes OM protein, lipids, nucleic acid, and so on.

Phospholipids and lipopolysaccharides are vital elements in maintaining the morphology and stability of the vesicles. Lipid rafts, composed of sphingomyelin and cholesterol, communicate between vesicles and host cells [26]. The phospholipid bilayer comprises the inner membrane (IM), which confines the cell’s cytoplasm. Multiple IM proteins are combined with PG to link the two layers together. The accumulation of phospholipids leads to asymmetric dilation of the OM, which is the cause of OMV formation. The phospholipid bilayer forms the walls of OMVs, confining and protecting the cargo. In addition to its function of limitation, the phospholipid bilayer participates in important bacterial activities, such as energy generation [27].

OMPs, localized in the outer membrane of Gram-negative bacteria, are pivotal in cell invasion and apoptosis. Furthermore, they aid in *A. baumannii’s* survival in the host [28].

Outer membrane protein A (OmpA), a β-barrel porin with a 38 kD weight, is one of the most important proteins in the bacterial protein secretion systems. OmpA plays a key role in mediating attachment to host cells, autophagy, invasiveness, biofilm formation, and apoptosis of *A. baumannii* [29,30,31]. In its secretion procedure, the vast majority (>99%) of the amino acids in OmpA come from various clinical isolates of *A. baumannii* are retained, and they are of a different origin compared to the human proteome [30]. Therefore, it can be associated with the host’s immune response. In addition to the outer membrane, OmpA is also located in the mitochondria and induces apoptosis via a complex mechanism that involves the destabilization of mitochondrial membrane potential [32].

Omp33-36 is another outer membrane protein that acts as a channel for water. In addition to its transport function, its expression is also associated with adhesion, invasion, regulating autophagy, and the metabolic adaptability of *A. baumannii* [33]. In the same way as OmpA, Omp33-36 is highly conserved and has high immunogenicity in *A. baumannii*.

Outer membrane protein W (OmpW), an eight-stranded β-barrel porin protein, forms a channel through the outer membrane to absorb small hydrophobic molecules. OmpW is also involved in colistin binding and helps regulate bacterial iron homeostasis in *A. baumannii* [34].

Additionally, many outer membrane proteins act as a channel of nutrition, which is a process closely associated with cell survival. For example, the carbapenem-associated outer membrane protein [35] uptakes glycine and ornithine. Outer membrane carboxylate channels AB1 (OccAB1), or OprD, allow diffusion of essential amino acids into the cell [36,37].

Furthermore, outer membrane proteins are also involved in the antimicrobial resistance of *A. baumannii.* CarO, with the deletion of the gene encoding a 29 kDa protein-conferred resistance to imipenem, plays a vital role in carbapenem resistance [35]. The surrounding environment affects the protein components in OMVs. For instance, Sung et al. showed that antibiotic treatment could modulate proteome components in *A. baumannii* OMVs [38]. OMVs also carry virulence factors and toxins, including phospholipase C, superoxide dismutase, and catalase, which are related to host tissue adhesion and invasion [26]. In the throes of an attack from antibiotics, enzymes carried by OMVs can either bind to or absorb the antibiotics to avoid the drug’s attack (Table 1) [38].

LPS, an essential component of OMVs, comprises lipid A, core polysaccharides, and O antigen. It plays an essential role in the biogenetic, virulence, and antibiotic resistance of OMVs. LPS participates in the formation of OMVs through electric change and modification [19]. They are also potent activators of immune cells and can be specifically recognized by the Toll-like receptor 4 (TLR4) receptor. Based on these characteristics, LPS can be applied to increase the yield of OMVs and enhance the immunogenicity of OMV vaccines. However, the immunogenicity of LPS is also a problem for OMV-based vaccines because it can initiate an overreaction and, as a result, cause unnecessary damage to the host. LPS are the primary target of polymyxins. Therefore, LPS modifications, such as modification of lipid A with phosphoryl ethanolamine and 4-amino-4-deoxy-L-arabinose, protect bacteria from polymyxin antibiotics [39].

Almost all Gram-negative bacteria can secrete OMVs, but the vesicle composition varies. It is demonstrated that the biogenesis and composition of OMVs are linked processes. The function of OMVs depends mainly on their cellular components and biological activity. For example, *S. Typhimurium* provokes OMV biogenesis to shed unfavorable LPS and proteins as an adaptation to new environments, which causes the enrichment of LPS and OM proteins in its OMV [40]. OMVs from commensal *Bacteroides* are enriched with glycoside hydrolases in order to degrade environmental polysaccharides. Therefore, the composition of OMVs can reflect their functions, which can help in the development of OMV-based vaccines for different bacteria.

The nucleic acid carried by the OMVs includes DNA, mRNA, miRNA, and other non-coding RNA [41]. In summary, OMVs contain signaling molecules that influence cellular communication, host immune responses, and provide nutrition for bacteria [19]. These components are also the biological basis of OMVs as possible subunit vaccine candidates, guiding vaccine preparation.

**Table 1 vaccines-12-00049-t001:** Virulence factors and mechanisms of OMVs in *A. baumannii*.

Virulence Factors			Mechanisms	Reference
Proteins	Outer membrane proteins	OmpA	Mediating attachment to host cells via fibronectin Invading epithelial cells Biofilm formation Apoptosis Associated host immune response of *A. baumannii* Regulating autophagy Transition from biofilm formation to maturation	[29,30,31,42,43,44,45]
		CarO	Uptake of glycine and ornithine Carbapenem resistance Adhesion and virulence in host cells via inhibition of NF-kβ signaling	[35,46,47]
		OprD/OccAB1	Allowing diffusion of essential amino acids into the cell Combating host-induced nutritional immunity and stress survival	[36,37]
		Omp33-36	Inducing apoptosis in host cells Regulating autophagy Adhesion and invasion Cytotoxicity and metabolic adaptability of *A. baumannii.*	[33,48,49]
		DcaP	Biofilm formation Uptake of clinically relevant negatively charged β-lactamase inhibitors	[50,51]
		OmpW	Colistin-binding site Absorbing small hydrophobic molecules	[34,52]
	Fimbrial proteins	CsuA/BABCDE	Attachment to and formation of biofilms on abiotic surfaces Transition from biofilm formation to maturation	[31,53]
		ABAYE2132	Adhesion and invasion Biofilm formation Involved in the motility of *A. baumannii.*	[54]
	Phospholipase	Phospholipase D	Hydrolyzing the phosphodiester bond Catalyzing transphosphatidylation reactions Invading epithelial cells	[54,55]
		Phospholipase C	Forming lipid rafts with cholesterol Cellular damage	[56]
	Other types of proteins	Iron acquisition system	Helping bacterial growth under iron-deficient conditions Regulating the survival of *A. baumannii* in the cytoplasm	[57,58]
		AbaI autoinducer synthase	Normal biofilm development Later stages of biofilm maturation Surface-related motility Drug resistance Invasion into epithelial cells	[59,60]
		Biofilm-associated protein (Bap)	Biofilm formation Intercellular adhesion in mature biofilms	[31,61,62]
		BfmRS	Biofilm formation on abiotic surfaces Transition from biofilm formation to maturation	[31,63]
		Penicillin-binding protein 7/8 (PBP-7/8)	Cell wall remodeling Directly or indirectly affecting serum drug resistance	[63,64]
		PNAG-constituted biofilm	Biofilm formation Drug resistance Maintaining the integrity of *A. baumannii* biofilms	[65,66]
Polysaccharide	Capsular polysaccharide (CPS)		Protecting bacteria from environmental damage Participating in host cell interactions Protecting phagocytosis and complement-mediated bactericidal effects	[67,68,69,70]
	Lipopolysaccharide (LPS)		Core oligosaccharide glycosyl transferase Resistance to normal human serum Conferring a competitive advantage for survival in vivo Drug resistance	[39,71]

### 2.2. Preparation Methods

OMVs are non-living particles. They can be separated from bacterial cells or synthesized by artificial methods. Various methods have been developed for preparing *A. baumannii* OM vesicles (AbOMVs) (Figure 2).

Spontaneously released OMVs (sOMVs) are isolated from supernatants by ultracentrifugation and filtration and are composed of a high percentage of periplasm proteins. The first step in the growth of bacteria, followed by centrifugation and ultrafiltration. To pursue higher purity, density gradient separation can further separate OMVs. The high lipid content of vesicles determines their low-density properties. Based on this characteristic, density gradient centrifugation effectively separates OMVs from other dense substances [72]. Compared with OMVs prepared in other ways, sOMVs are more likely to harbor fewer membrane-anchoring proteins and lytic transglycosylases while containing more periplasm proteins and molecule transporters [73].

Obtaining OMVs through ultrasonic bacterial culture is another method. Ultrasound can increase membrane instability, resulting in the rupture of bacteria. *A. baumannii* cells were disrupted by EDTA/lysozyme treatment after first using sonication. Sucrose-extracted OMVs (SuOMVs) were then separated from the total protein by sucrose gradient isopycnic centrifugation. SuOMVs have several advantages as a viable type for use in OMV vaccines. The relatively small size of SuOMVs enables effective internalization by APCs. They also contain a higher percentage of inner membrane proteins and residual DNA. It is demonstrated that mice immunized with suOMVs had the highest IgG titers and 7-day survival rate, followed by sOMVs and native OMVs [72]. Nevertheless, the reduction in bacterial burdens is lower in the suOMVs group 24 h post-sublethal challenge.

Native OMVs (nOMVs) were obtained from bacterial cells cultured in 10 L of LB broth. After the suspension, the living bacteria were cut with a high-speed disperser and were collected via differential centrifugation [15]. Finally, *A. baumannii* nOMVs were washed and diluted in a PBS solution. Among the three AbOMVs mentioned above, nOMVs demonstrated the highest level of LPS, and their production level was too low for application to vaccine manufacture. nOMVs have a relatively prominent advantage in inducing cytokine secretion, which is often related to a robust immune response.

Detergent deoxycholate was a valuable method for extracting OMVs from bacteria, thus solving the low yield problem. OMVs prepared in this extraction method were called detergent OMVs (dOMVs). As well as the high yield, using detergent effectively decreased the level of LPS and endotoxicity in the OMVs. Also, due to the reduction in LPS, dOMVs have to sacrifice a certain degree of their immunogenicity [74].

Vaccines based on bacterial biomimetic vesicles (BBVs) have seen robust protection against *Klebsiella* spp. The primary mechanism of BBV technology is high mechanical pressure. The bacterial membrane was first softened, and then the bacteria were passed through a gap to form buds through a high-pressure homogenizer under different pressure strengths. In this process, intracellular proteins and nucleic acids were released, and bacterial membranes self-assembled into vesicles. Intracellular proteins enveloped in OMVs were sharply decreased while the percentage of membrane proteins increased. The outcome of the research suggested the abundant advantages of BBVs as vaccines. BBVs were efficiently taken up and processed by dendritic cells and could induce bacteria-specific humoral and cellular immune responses in the body. Moreover, BBVs have shown their potential to defend against antibiotic-resistant bacteria [75] and important breakthroughs in vesicle preparation as a novel vesicle production formation. In the case of *K. pneumonia*, the yield of BBVs is high, and its production process can be artificially controlled. Their safety is a prominent feature because BBVs mainly consist of outer membrane proteins and all bacterial nucleic acid components are removed. Therefore, applying BBV technology to prepare *A. baumannii* OMVs may provide similar benefits, and further exploration is desired.

Genetic engineering has also been used to produce modified OMVs. OMVs can be artificially manufactured through genetic recombination to carry specific proteins or polysaccharides [76,77]. Moreover, deleting or silencing the gene relating to the OmpA, the VacJ/Yrb transporter system and others controlling the production of the OMVs is an effective tool to enhance the efficacy and yield of OMVs [18,78]. Compared with the wild-type strain, the ΔbfmS and ΔzrlA mutants secreted more OMVs and released more proteins via OMVs in the supernatants [79,80]. The specificity of genetic engineering vaccines is potent, but at the same time, their production processes are complex, which is not conducive to the widespread manufacture of OMV-based vaccines.

Overall, *A. baumannii* OMVs obtained through different methods have specific features, among which SuOMVs show apparent advantages, including strong immunogenicity and suitable vesicle size. Furthermore, novel vesicle preparations such as BBVs have shown important research breakthroughs and offer an alternative to the preparation of *A. baumannii* OMVs. The production of vesicles is an essential step in the research into safe and effective vaccines. Improving its immune-protective effect reduces its toxicity, and calculating production cost is an additional important topic.

## 3. The Role of OMVs in the Pathogenesis of *A. baumannii*

### 3.1. Virulence

Various components of bacterial virulence are required for the pathogenesis of pathogens. The natural OMV release is a novel pattern for *A. baumannii* to exert its virulence over a long distance and to expand its range of virulence [81]. Virulence factors included in OMVs can be divided into two parts: (1) the presentation by OMVs of multiple copies of antigens to the immune system in their natural formation and orientation; (2) the co-delivery in OMVs of pathogen-associated molecular patterns (PAMPs) [81]. These factors are related to the regulation of host immunity, adherence, and antibiotic resistance. OMV can be phagocytized and processed by antigen-presenting cells (APCs). The OMV-delivered antigens are then presented by APCs to CD4+ T cells, leading to the generation of antigen-specific B-cell responses.

Polysaccharides are a crucial toxic component of Gram-negative bacteria, which is also enriched in OMVs. There are two types of polysaccharides in *A. baumannii* OMVs that contribute to the survival and invasion of the bacteria. Capsular polysaccharides provide protection against phagocytosis-complement-mediated bactericidal effects and participate in host–cell interactions [67]. LPS assists in the survival and pathogenesis of *A. baumannii* in a TLR4-dependent manner [82]. Recognition by the TLR4/MD2 receptor on the cells provokes NF-κB- and IRF3-dependent gene expression. The lack of LPS causes lower bacterial loads in post-infection tissues and significantly reduced levels of the inflammatory cytokines IL-1b, TNF-a, and IL-6 in a mouse model of disseminated *A. baumannii* infection [83]. Both polysaccharides are significant virulence determinants, resulting in the powerful pathogenicity of OMVs.

Membrane proteins in OMVs are responsible for eliciting a potent innate immune response [84]. Multiple OMPs anchor to other OMVs, contributing to adhesion, invasion, and nutrient achievement. OmpA in *A. baumannii* OMVs can provoke an intense pro-inflammatory response in macrophages and increase cell death [85]. The expression of IL-6 and TNF-a significantly increases with OmpA, which subsequently results in a more potent inflammatory pathological change in mouse lung tissue [86]. When treated with antibiotics, OmpA-immunoreactive components in the OMVs increased, enhancing the virulence of the biofilm [87]. The OM proteins in OMV can also enhance this binding function. For instance, Omp33 can induce bacterial adhesion through interaction with fibronectin [33,84], which is the basis of colonization and invasion. Other proteins, including fimbrial and phospholipases, have demonstrated their ability to form biofilm, resist extreme environments, etc. [31,54].

Thioredoxin A protein (TrxA) is a virulence factor, playing an important role in resistance to oxidative stress. It facilitates host immune evasion partly through the alteration of type IV pili and cell surface hydrophobicity. TrxA-deficient (DtrxA) *A. baumannii* OMVs provoked more J774 macrophage-like cell deaths and increased lung permeability [88].

In addition to the main virulence factors mentioned above, a number of bioactive molecules are embedded in OMVs, and the biological activities they mediate are yet to be explored. The study of the quantitative relationship between the virulence determinants found in OMVs and their immunological effects can more clearly enlighten the biological function of OMVs.

### 3.2. Pathogenicity

*A. baumannii* is closely related to plenty of nosocomial infections, including pneumonia, bloodstream infections, meningitis, and skin and urinary tract infections [89]. It is believed that overexuberant inflammatory responses to infectious pathogens are the key to damage to the tissues involved. OMVs play a vital role in the pathogenesis of bacteria by imparting virulence factors and regulating their immune responses [90]. In addition, *A. baumannii* OMVs also assist the bacteria in gaining robust adaptability to living environments and increasing their tolerance to antibiotics.

Biofilm is a bacterial community structure protecting bacteria from unfavorable environmental factors, and OMVs appear to play a structural role within these communities. Proteins involved in OMVs can help form biofilms, improving the survival of bacteria on the surface of medical devices, such as ventilators in the ICU. Therefore, the existence of OMVs improves bacterial resistance and thus enhances bacteria’s pathogenicity [25].

It is widely believed that OMVs enriched with antibacterial components can facilitate entry into nonphagocytic cells via fusion or an endocytic pathway [91] and result in death. With the interactions with lipid rafts, the fusion of OMVs can facilitate the delivery of toxins into host cells. The endocytic route is regulated by clathrin or caveolin [27,92]. The attack of bacteria can raise the epithelial permeability, which weakens the barrier integrity. Epithelial cell dysfunction is a critical pathophysiological process in pneumonia and is linked to invasive infections [93]. As a long-distance delivery system, OMVs enriched with virulence molecules enter the host cells and kickstart the invasion. Various OM proteins in OMVs mediate the binding of bacteria and host cells, increasing the surface expression of toll-like receptor 2 (TLR2) and activating the downstream signal conduction to mitochondrial collapse and nuclear pyknosis [86,88]. Furthermore, OmpA can also induce mitochondrial fragmentation in host epithelial cells and alveolar macrophages, as seen in a mouse lung infection model, by activating the host GTPase dynamin-related protein 1 (DRP1). Macrophages may detect the mitochondrial stress caused by OMV exposure, which then provokes apoptotic cell death factors [94,95].

Numerous factors can influence the pathogenicity of OMVs. Upon analyzing the lung histology of mice stimulated with *A. baumannii* OMVs, it was found that OMVs can mediate pulmonary consolidation by neutrophil recruitment. Furthermore, wild-type mice had more severe pulmonary consolidation than TLR2-deficient and TLR4-deficient mice. Mouse lungs treated with OMVs exhibited raised mitogen-activated protein kinase-related phosphorylated P42/44 (or Erk1/2) and pro-inflammatory cytokines including IL-1, IL-6, and TNF-α [96].

Iron is a significant factor in the survival of many organisms. By affecting other virulence factors such as cell motility, cell adherence, and biofilm formation, iron acquisition pathways contribute to the invasion of the bacteria. Therefore, TonRs, which are responsible for taking iron, have been confirmed as essential elements in the pathogenicity of OMVs [97,98].

OMV-induced pathogenesis is also related to drug resistance. OMVs harbor resistance determinants, either a resistance gene or a degradative enzyme against relative drugs, diluting the detrimental effect on the membrane. Small RNAs (sRNAs) packed into OMVs can be stably transported either in OMVs or directly absorbed into the host cells [99]. It is confirmed that sRNAs can contribute to *A. baumannii* antibiotic resistance and pathogenesis [100]. However, the specific mechanism of this procedure needs further study.

OMVs serve as contributors to creating a suitable microenvironment for the survival and invasion of the pathogen in a host. Therefore, they play a vital role in enhancing the pathogenicity of *A. baumannii*, including expanding the bacterial invasion range and resisting any host attack.

### 3.3. The Role of Bacterial Drug Resistance

In the absence of any effective intervention for *A. baumannii* infection, a large number of antibiotics have been applied in clinical use. Excessive and combined use of antibiotics can accelerate bacterial resistance. The abuse of antibiotics leads to an increasing mutation of *A. baumannii* [15]. This results in resistance to many antibiotics, such as carbapenems, tigecycline, etc. [101]. The current severe situation in *A. baumannii* infections is that no effective drugs are available [102]. Multiple mechanisms participate in the bacterial drug resistance of *A. baumannii*, which includes enzymatic inactivation of antibiotics, target mutation, and reduction in OM transportation [103].

Furthermore, *A. baumannii* can form biofilms and thus improve its survival on the surface of medical devices. However, the relationship between biofilm formation and antibiotic resistance is still unclear. Previous studies have proved that OMVs produced by *A. baumannii* contain antibiotic-resistance factors, such as AmpC and OVAs [104], which can strengthen the occurrence of bacterial drug resistance through plasmid delivery and other pathways [105].

Enzymatic modification is one of the most common mechanisms of antibiotic resistance. Bacteria produce enzymes that bind and degrade antibiotics to attain drug resistance. For *A. baumannii*, these enzymes are mainly monooxygenases for tetracyclines and beta-lactamases for beta-lactams. Beta-lactamase catalyzes the hydrolysis towards antibiotics like β-lactam antibiotics and can be developed in OMVs. Among the metallo-β-lactamases enriched in OMVs, NDM-1 is selectively secreted into *A. baumannii* and exhibits high carbapenemase activity [106]. However, the resistance to carbapenems has dramatically increased over recent years among Gram-negative bacteria [107]. Research has demonstrated that *Acinetobacter* OMVs can reduce extracellular β-lactam antibiotic concentrations through the action of both carbapenem-hydrolyzing class D β-lactamases and AmpC-like β-lactamases [108,109]. Bacterial genes influence the level of β-Lactamase. The insertion sequence IS*Aba1* located upstream of the *bla*_ampC_ gene can promote the expression of β-Lactamase, thereby enhancing resistance to third-generation cephalosporins.

Antibiotics need channels or transporters to enter the infected cell. Therefore, factors changing the outer membrane’s permeability can influence the antibiotics’ effect. Porins and pumps are the two main mechanisms that affect transportation through membranes. OmpA, a non-specific porin in *A. baumannii,* has an N-terminus transmembrane β-barrel domain. It plays a structural role in antimicrobial resistance [28,110]. It is speculated that OmpA is involved in transporting antibiotics out of the periplasmatic space.

Additionally, in mutant strains lacking OmpA, the outer membrane becomes unstable, and susceptibility to antibiotics increases [111]. It has been demonstrated that CarO acts as an uptake channel for L-ornithine and possibly also for carbapenems [112]. The function of other OMPs in decreasing the permeability of the outer membrane has also been studied [113]. In addition to porins, the efflux pump plays another vital role in decreasing the transition of the antibiotics into cells. The major facilitator superfamily (MFS), the resistance-nodulation cell division (RND) family, the small multidrug resistance (SMR) family, and the multidrug and toxic compound extrusion (MATE) family are four classes of efflux pumps that are associated with *A. baumannii* antimicrobial resistance. Pumps expel antibiotics out of the cell, which decreases drug accumulation and increases the minimum inhibitory concentration (MIC) required. Efflux pumps could also be implicated in the early stages of infection, causing complications such as adhesion to host cells and colonization [101].

Target site alternation is the third main mechanism of antibiotic resistance in *A. baumannii,* and it is generally associated with nucleic acids. Through the mutation of type II topoisomerases, infected cells can escape the interference of quinolones and thus induce supercoiling in collaboration with DNA nucleases. Target mutations of rRNA methylase genes and RNA elongation inhabitation play a structural role in preventing the attack of aminoglycosides. Additionally, genes conferring resistance can be transported by means of integrons, gene cassettes, transposons, and conjugated elements [113].

OMVs that are secreted in the periplasm serve as decoys for the binding of the membrane-targeting antibiotics [114]. The function of off-targeting has been demonstrated in *E*. *coli*, *P. syringae*, and other bacteria.

OMVs secreted by *Acinetobacter* can serve as delivery compartments, transferring a carbapenemase-containing plasmid to other *Acinetobacter* spp. [115]. For example, *A. baumannii* OMVs can transfer *bla*_OXA-24_, fastening the dissemination of carbapenemase genes in *A. baumannii* [116]. Moreover, the secretion of NDM-1 into OMVs mentioned before is proposed as a way that may help disseminate the *bla*_NDM_ gene [106]. Horizontal gene transfer is an essential link in acquiring extensive drug resistance. In addition to the function of transmission, OMVs can prevent the resistance genes from being degraded by nuclease [105].

An in vitro human microbiota experiment showed that the OMVs released by *A. baumannii* act as bait and then protect the actual bacterial community when under the treatment of polymyxin B. This finding further confirms the critical role of vesicles in inducing antibiotic resistance. On the other hand, antibiotics can promote OMV secretion and modulate OMV protein components, resulting in increased pathogenicity towards the host cells [38].

The rapid increase in antibiotic resistance in the last few decades shows the remarkable genomic plasticity of *A. baumannii* and, as a result, the available treatments for *A. baumannii* infections are increasingly becoming limited due to its increasing resistance to antibiotics [117]. OMVs may be a breakthrough in this dilemma of antibiotic resistance.

## 4. Application Progress of *A. baumannii* OMVs in Vaccine Development

### 4.1. Immunogenicity and Immunoprotective Effect

*A. baumannii* is a common cause of hospital-acquired pneumonia, especially in Asia, resulting in very significant loss of life, health-care costs, and an increasing number of vulnerable patients [3]. Strong pathogenicity and increasing antimicrobial resistance may be responsible for these conditions. The composition of most outer membrane vesicles is conserved and recognized by the body as antigens, so OMVs, as a complex of multiple virulence factors, are endowed with strong immunogenicity. In addition, OMVs serve as a platform for other virulence factors. Once OMVs enter the host, neutrophils, and macrophages are recruited, and they elicit the release of corresponding inflammatory cytokines. Several components in OMVs are PAMPs, which can be sensed by PRRs such as TLRs, thus provoking the inflammatory response [14]. A significant role in recognizing OMV PAMPs has been attributed to TLR4, which is involved in recognizing LPS. Nevertheless, OMV also contains other PAMPs. TLR2 has also been shown to participate in the recognition of bacterial lipoproteins. With the help of APCs, the OMV-delivered antigens are then presented to CD4+ T cells. B cells participate in the immune response successively, and protective antibodies such as IgA and IgG are released [118]. The contact between *A. baumannii* and the body’s internal environment can trigger diverse immunoprotective methods to protect the host. The number of OMVs released rapidly increases when *A. baumannii* is exposed to pleural fluid, suggesting that OMV release may be a strategy used by *A. baumannii* to respond to the stress caused by pleural fluid [119]. Plenty of studies demonstrate that pre-administration of vesicle stimulation can effectively limit the bacterial load and inflammatory cell infiltration in the lung, and the survival rate of pneumonia mice is significantly improved [120]. Memory B cells can respond quickly by producing antigen-specific antibodies to protect the body the next time bacteria invade. Once *A. baumannii* invades the body again, memory T cells and B cells could induce rapid and effective immune responses. Furthermore, OMVs from various other respiratory pathogens have been shown to induce cytokine release, including IL-1, IL-6, and IL-8, which are involved in activating immune response and invasion.

In addition to the robust immunogenicity of OMVs, one study shows that antibodies against OMVs can vastly increase the susceptibility of the bacteria to quinolone antibiotics. Combining anti-AbOMV antibodies with antibiotics reduced the bacterial load in the organs, resulting in a higher survival rate [121].

Immunogenicity is the key characteristic of bacterial OMVs, manifested by stimulating an innate immune response and eliciting antigen-specific antibodies. The OMV-mediated immune response is the cumulative result of multiple antigen activations. Hence, it is not surprising that OMVs are regarded as hopeful candidates for subunit vaccines against *A. baumannii*.

### 4.2. Mechanism of Immune Response Mediated by OMVs

Epithelial cells are the first barrier for bacteria in their invasion of the host. OMVs can enter host cells via diverse mechanisms, including micropinocytosis [92], endocytosis, etc. Once they enter the host, they will stimulate the non-specific immunity of the host. Innate immune cells, including neutrophils and macrophages, will be activated to elicit inflammatory responses against the OMVs (Figure 3) [122].

Activated neutrophils produce numerous cytokines and chemokines to protect the host. On the other hand, some virulence factors carried by OMVs can prevent the antimicrobial activity of neutrophils and, hence, contribute to a decrease in the release of cytokines that have been simulated in the host by OMVs [123].

OMVs are phagocytosed by macrophages, which contributes to the secretion of TNFα IL-8 through NF-κB activation [124]. At the same time, macrophages, as antigen-presenting cells, can stimulate T cells to elicit adaptive immune responses. Nevertheless, a study has shown that bacterial OMVs can change the metabolism of macrophages, causing pyroptosis and apoptosis [122]. The dysfunction of and reduction in the immune cells are vital to the disease. Furthermore, the production of anti-inflammatory cytokines, such as IL-10 and IFN-α, is diminished, as reported [125]. Therefore, in the stage of innate immunity, neutrophils and macrophages provoke the immune response and elicit immune evasion as well.

PRRs are membrane-bound proteins chiefly expressed by innate immune cells, which play a critical role in the host’s innate immunity by recognizing PAMPs. TLRs are one of the significant PRRs in Gram-negative bacteria, including *A. baumannii* [14]. For example, the interaction of TLR4 and lipid A can induce inflammatory cytokine expression and subsequent neutrophil recruitment [126]. The activation of TLR assists in the stimulation of antigen-presenting cells and the recruitment of neutrophils [96], which is vital to the innate immune response against *A. baumannii* [114,123].

The antigen-presenting of APCs is the first step of adaptive immune response. It is confirmed that in mice injected with *A. baumannii* OMVs, bone marrow-derived dendritic cells (BMDC) expressed more CD80, CD86, and MHC-II [125]. Secondly, APCs present OMV antigens to CD4+ T cells and then promote T helper cells (Th) to polarize into different subtypes. In this process, OMVs may play a role as adjuvants, which contributes to the T cells’ cross-priming [127]. With the assistance of Th cells, B cells are activated to secrete antibodies. Plasma cells release a large amount of IgG and IgM, as shown in Figure 3, and secretory IgA surges, especially in a protective humoral response against mucosal infection [128]. In this way, OMVs endow the host with targeted immune protection. When the bacteria invade again, the immune system can react effectively and reduce host cell damage.

Consequently, OMVs contribute to innate immunity by activating the neutrophils and macrophages. Then, they contribute to an adaptive immune response in the host by enhancing the antigen-presenting ability and stimulating both humoral immunity and cellular immunity.

### 4.3. Artificial Biomimetic OMVs in Vaccine Development

A vaccine is composed of virulence components derived from pathogens. Therefore, it can stimulate an adaptive immune response in an antigen-specific manner [129]. Its lasting and effective antibody protection can provide higher clinical benefits than antibacterial drugs and symptomatic treatment. Surface-exposed membrane antigens and the preservation of good physico-chemical stability are key features of OMV vaccines. Therefore, vaccines are promising alternative strategies to control *A. baumannii* infections. Subunit vaccines, in particular, have the advantages of purity, stability, and efficiency. OMVs are of particular interest among the subunit vaccine candidates due to their immunogenicity and association with bacterial drug resistance [13,26]. Compared with the whole-cell vaccine, OMVs have unique traits. First, the use of inactivating agents can be skipped. Therefore, PAMPs can remain in their native states [77]. Furthermore, their small diameter enables OMVs to enter the lymphatic system easily and even cross the blood–brain barrier. The characteristics of OMVs that trigger a potent immune response have already been demonstrated with several OMV-based vaccines. In 1987, the OMV-based vaccine against Neisseria meningitis was first licensed for use. Subsequently, OMVs have been successfully employed as vaccines to prevent meningococcal group B outbreaks [130]. The efficiency of these vaccines has been estimated to be at least 70% in New Zealand, 83% in Cuba, and up to 87% in Norway. Additionally, OMV-based vaccines against N. meningitidis serogroup B in New Zealand have been found to also reduce the rate of gonorrhea [12,131].

OMVs are known to be highly immunogenic and are a sufficient candidate for effective vaccines. The characteristics of their nanostructure can provide higher stability for the virulence factors carried by OMVs. The adjuvant activity of OMVs has been investigated by combining them with antigens. For example, recombinant Omp22-OMV vaccines protected mice from *A. baumannii* in a murine sepsis model [14], which suggests that OMVs may act as a novel antigen delivery carrier in vaccine development. OMV vaccines can also be decorated with specific proteins or polysaccharides from heterologous pathogens. This has been demonstrated by incorporating heterologously expressed outer membrane and periplasmic proteins into OMVs produced by the laboratory and pathogenic *E. coli* strains [132]. In addition to their simple loading function, engineered OMVs effectively improve the body’s immune response. Immunization with *E. coli* OMVs expressing *A. baumannii* Omp22 stimulated higher levels of Omp22-specific antibodies than immunization with a recombinant Omp22 protein formulated with alum [14]. OMV decorated with heterologous antigens has been shown to promote enhanced immunogenicity compared to traditional formulations. With the assistance of bacterial engineering, such as recombinant DNA technology, recombinant OMV-based vaccines that embed antigens from different pathogens can be created.

Additionally, OMVs have also been found to have great potential for loading antibacterial drugs and introducing the drugs into bacteria [133]. Nevertheless, this method was limited to certain antibiotics, such as quinolone antibiotics. In conclusion, OMVs are employed to stimulate the immune response and as vehicles for specifically expressed antigens.

Regarding preparation, OMVs are non-living particles that can be easily separated and purified from bacteria. As an adjuvant, OMVs can be decorated with specific proteins or polysaccharides from heterologous pathogens [81]. Tailored OMVs can then be produced through advanced technology [134]. Irfan et al. found that the translucent subpopulations of *A. baumannii* could enhance the production and immunogenicity of OMVs [135]. By comparing the immunoglobulin levels, agglutination rate, and survival rate, a study has demonstrated that OMVs isolated from a clinical strain can elicit a more potent Th2 response and immune protection [125]. Therefore, the features and components of the OMVs from different colony variants need further study, which may provide a new strategy for the preparation of OMVs. Other routes of vaccination result in diverse effects of immune protection and OMVs have been proven to be immunogenic in all ways [8]. Compared with intramuscularly immunized mice, subcutaneously immunized mice have lower lung inflammation, a lower bacterial load, and better protective effects. This consequence may be associated with increased levels of IgG1 by enhancing the Th2 response. Intranasal vaccination with OMVs showed the most robust protection against infection with *A. baumannii*. Intranasally delivered OMVs induced an OMV-specific IgA response, which further enhanced protection against the bacteria [136]. The outcome shed light on a new method to improve the local mucosal and systemic immune responses of OMV-based vaccines through mucosal vaccinations.

Despite the great potential that OMVs have shown in the antibacterial area, many problems still need to be solved before clinical use. The main concern is the safety of the vaccine. Lipid A is an endotoxic component of LPS in OMV and can lead to intense, even lethal, responses in the host. In nOMV vaccines, high potential reactogenicity exists due to unmodified PAMPs. Since the majority of the target group infected with *A. baumannii* are frail patients, the reactogenicity may be fatal to them. Due to the possibility of contamination with pyrogenic LPS during the preparation of vaccines, the large amount of LPS may impede the application of OMV vaccines [13,82]. Several methods have been developed to reduce the LPS toxicity of OMVs. The first measure is to reduce LPS levels by treating purified OMVs with detergents. However, in the purification procedure, lipoproteins, which act as TLR agonists, were lost and thus reduced the adjuvant properties of OMVs. The second strategy is the low expression of LPS-related genes. The LPS-deficient inactivated whole cell was developed by deleting the *lpxD* gene essential for LPS, which was confirmed to elicit immune responses similar to those of wild-type *A. baumannii* [83]. Reduced endotoxicity can also be achieved by modification of lipid A structure or inactivation of the genes encoding lipid A acyltransferases, resulting in reduced stimulation of TLR4. Moreover, the study shows that increasing the dose of LPS-free OMVs results in powerful protection from the infection [137]. Additionally, the OMV content varies in different strains, which may be a barrier to the efficacy of specific strains. The secretion level of OMV in many strains is low, which means that large volumes of pathogens or purification methods are needed to increase the efficacy of the OMV-based vaccine.

*A. baumannii*, as an opportunistic pathogen, has caused and continues to cause severe harm to vulnerable patients and has a significant impact on global public health. Vaccines are practical tools to prevent and control *A. baumannii* infection and colonization. Compared to other control strategies and regular vaccines, OMV vaccines offer a better alternative for people at high risk of infection. Due to their reduced components, OMV vaccines have less toxicity and side effects than traditional ones, which is essential for patients who are physically weak. Furthermore, OMVs are secreted from the bacteria, and many components are highly conserved. This characteristic can strongly induce cellular and humoral immunity, effectively preventing severe infections and bacterial colonization. Under the premise of effectively reducing the occurrence of infection, the health-care costs, for both individuals and society, would also be reduced. OMVs represent an attractive and innovative potential for developing safe and potent vaccines against pathogens.

## 5. Conclusions

In the face of multi-drug-resistant, or even pan-drug-resistant, and highly pathogenic *A. baumannii*, there is an urgent need for solid infection control measures to address the problem at its origin. Vaccines are a robust preventative and control measure to reduce microbial infections and pathogenic damage to the organism. Today, vaccines are also being widely studied for noninfectious diseases such as cancer.

The ability of OMV to induce a durable antitumor immune response has been demonstrated in multiple tumor models [138,139]. The simplification and recombination of antigenic components have received significant attention for improving the efficacy of vaccines and reducing production costs. However, no *A. baumannii* subunit vaccine candidate has yet to be applied in the clinic. This may be related to the weaker immunogenicity and easier degradation of subunit vaccines than in traditional vaccines. To solve this problem, we should suggest that the focus should be on the general study of candidates and take full advantage of their specific features. The feasibility of vaccine development for *A. baumannii* OMVs, where OMVs can act as both antigens and adjuvants, has been verified in several studies.

The immunoprotective effects of OMVs have been repeatedly demonstrated in animal models. In addition, with the development of multidisciplinary emerging technologies, we can improve the impact of immune response and protection through biological techniques such as the extraction of antigen epitopes and genetic engineering [139]. OMV vesicle vaccines have been applied in clinical practice and have achieved good clinical benefits, showing us the OMV vaccine’s infinite potential. The adjuvant function of OMV brings more possibilities for vaccine development. As a carrier and reaction enhancer, OMV is expected to combine with various antigenic components to improve the efficacy of the original vaccine and further reduce the production cost, which is very beneficial for its clinical development. Nevertheless, OMV vaccines against *A. baumannii* still face many problems, including the proportion of virulence elements [123]. Therefore, in the process of OMV vaccine development, how to attenuate unnecessary toxins and ensure the safety of the vaccine needs to be further studied.

Appropriate immunization methods can enhance the immune effect of subunit vaccines, and selecting an appropriate immunization method is essential for subunit vaccine research. The OMV-based vaccine helps packaged antigens resist degradation and destruction of the surroundings, and OMVs themselves can induce an immune response. Therefore, on the basis of existing studies, we can compare the immunogenicity and toxicity of subunit vaccines under different administration routes to find the best clinical choice. Furthermore, the OMV package and function depend on the surrounding environment the bacterium is exposed to, and the mechanism of how the OMV component shifts in response to a change in the environment needs to be addressed to assign specific functions to OMVs.

OMV vaccines against *meningococcal* disease have achieved effective and safe clinical application. There is a significant chance that OMV vaccines, not only against *A. baumannii* but also against other Gram-negative and viral pathogens, will be clinically available in the near future, for the benefit of patients and society.

## Figures and Tables

**Figure 1 vaccines-12-00049-f001:**
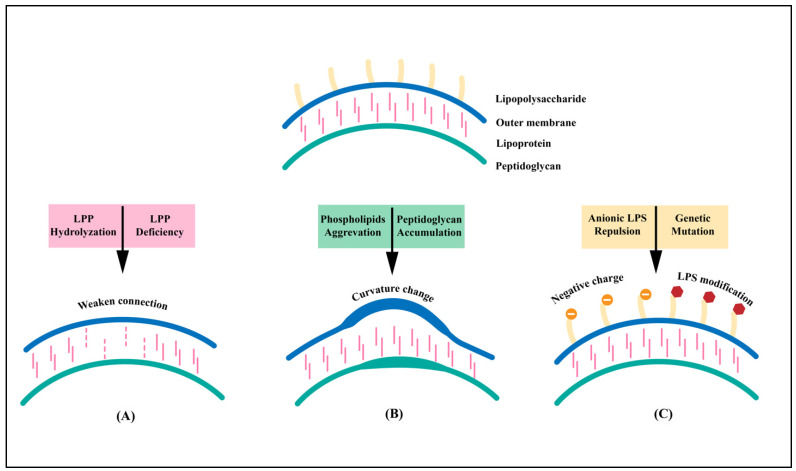
Schematic diagram of the biogenesis of *A. baumannii* OMVs. There are three basic models for the mechanism of biogenesis of *A. baumannii* OMVs. (**A**) LPP-PG covalently connects the bacterial OM and the peptidoglycan layer. When LPP is hydrolyzed, reduced, or even absent, the growth rate of the OM will exceed that of the peptidoglycan layer, resulting in OMVs. (**B**) The second model is based on the occurrence of curvature of the OM, which is associated with the accumulation of phospholipids and peptidoglycans in the OM. (**C**) LPS is an important biological molecule on the OM of bacteria and is closely related to vesiculation. The refusion of anionic LPS and mutations in LPS-related genes affect the biogenesis of OMVs.

**Figure 2 vaccines-12-00049-f002:**
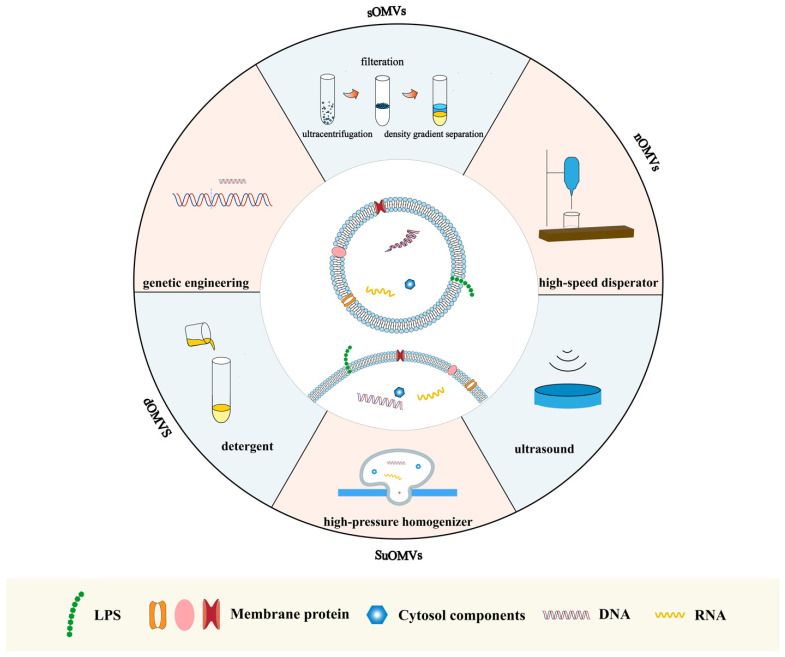
Schematic diagram of the artificial preparation of *A. baumannii* OMVs. Spontaneously released OMVs are isolated from supernatants by ultracentrifugation and filtration. Native AbOMVs are obtained by cutting living bacteria with a high-speed disperser. Detergent OMVs are obtained through detergent. OMV-prepared high-pressure homogenizers are called SuOMVs. OMVs can also be obtained using ultrasonic bacterial culture and genetic engineering. sOMVs spontaneously released OMVs; nOMVs, native OMVs; dOMVs, detergent OMVs; SuOMVs, sucrose-extracted OMVs.

**Figure 3 vaccines-12-00049-f003:**
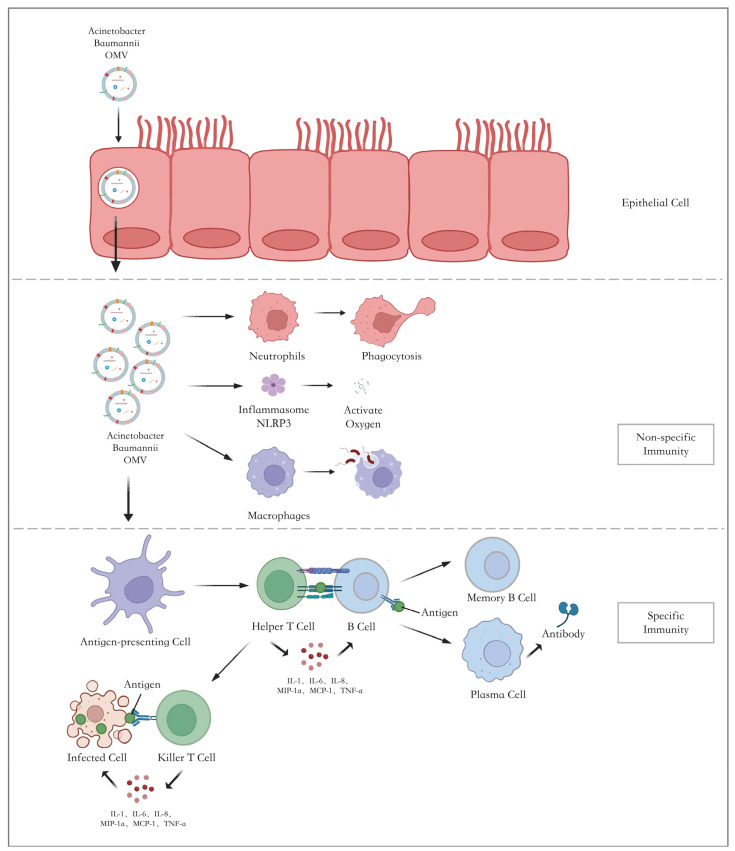
Schematic diagram of protective effect in *A. baumannii* OMV vaccine. *A. baumannii* OMVs enter the host through the epithelial cells and trigger non-specific immunity with the activation of neutrophils, macrophages, and inflammasome NLRP. The antigen is then recognized by antigen-presenting cells, which provokes the activation of humoral and cellular immunity. Antibodies produced by plasma cells neutralize the antigen. In parallel, killer T cells eliminate the infected cell directly or by releasing cytokines to induce the following kill. NLRP = Nod-like receptors; MCP = monocyte chemotactic protein; MIP = macrophage inflammatory protein; TNF = tumor necrosis factor.

## Data Availability

Data sharing is not applicable to this article.

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
