# Peer review of "Outer Membrane Vesicles from Acinetobacter baumannii: Biogenesis, Functions, and Vaccine Application"

_vaccines, 2023, doi:10.3390/vaccines12010049_

Round 1

Reviewer 1 Report

Comments and Suggestions for Authors

attached

Comments on the Quality of English Language

 Some minor editing is necessary to improve the clarity of the English language. 

Author Response

Response to Reviewer 1

Dear reviewer1,

Thank you for your patience on our submission, and we appreciate the comments concerning our manuscript. Those comments are all valuable for revising and improving our paper. We have studied comments carefully and have made correction which we hope meet with approval. All page numbers refer to the revised manuscript file with tracked changes. The corrections are highlighted with red color in the revised manuscript.

(1) In lines 457-460, we added detail on how antibiotic abuse has led to A. baumannii’s drug resistance and emphasized the study’s importance in the bacterial control.

(2) In lines 35-45, we have expanded on A. baumanniis inclusion in the WHO’s list of critical pathogens and highlighted its global health implications.

(3) In the part of the introduction (Page 2, Lines 46-65), we highlighted the urgent need for vaccines in the prevention and control of A. baumannii in the absence of effective intervention strategies by citing data.

(4) In lines 66-76, We added the definition of subunit vaccine and explained its advantages in purity, stability, and efficiency against A. baumannii.

(5) In lines 658-665, we introduced OMV-based vaccine in meningococcal disease and emphasized the potential of OMVs as vaccine candidates.

(6) In lines 84-93, We point out the gap this review aims to fill and hope that the generalization of OMVs can provide new ideas for developing other OMV vaccines.

(7) In the part of the preparation of OMVs, we added more details on the biogenesis process of OMVs. After the first paragraph of this section, we added a schematic diagram of the biogenesis of A. baumannii OMVs.

(8) In lines 106-110,124-145, 160-174, We elaborated on the mechanisms of OMV formation and cargo selection and pointed out the areas of uncertainty.

(9) In lines 148-159, We have offered more depth in how environmental conditions affect OMV production.

(10) In lines 246-255, we have discussed the composition and variation of OMVs in bacteria in depth.

(11) In the section on components(Pages 5-6 Lines 187-245), we added the introduction of the structure and function of important components.

(12) In lines 279-347, We described the existing OMV preparation methods in detail and compared them.

(13) In the section on the role of OMVs in the pathogenesis of A. baumannii (Pages 11-14, Lines 351-543), we elaborated on the virulence factors of OMVs, and we added a more detailed mechanism of how OMVs enhance A. baumannii’s virulence.

(14) In lines 473- 522, we discussed in detail the mechanisms of OMVs in enhancing antibiotic resistance.

(15) In the section on immunogenicity(Pages 14-15 Lines 544-585), we added the specific process of OMV PAMPs initiating the induction of immune response. We elaborated how OMVs increased the body’s response to a secondary bacterial attack.

(16) In lines 672-682, we added more details on OMVs' role in vaccine development.

(17) In lines 714-737, we focused on the current issues facing OMV-based vaccines, including their safety and efficacy.

(18) In the conclusion(Pages 19-20 Lines 752-800), We summarized the possible applications of the OMVs in A.baumannii and other pathogens and proposed some future research directions.

(19) The revised manuscript has been checked and edited by a professional editing company (MDPI).

Responses to each comment are listed as below:

This study aims to explore the biogenesis, functions, and potential vaccine applications of Acinetobacter baumannii outer membrane vesicles (OMVs). Key findings reveal OMVs’ role in pathogenesis and antibiotic resistance, highlighting their immunogenic properties and potential as vaccine candidates. The study suggests OMVs could offer a novel approach to combating A. baumannii infections. Although the paper presents a certain scientific interest, there are some important comments:

  • Please add more detail on how antibiotic abuse has led to baumannii’s drug resistance, which would provide context for the study’s importance.

R: Thank you for your suggestion. In the revised manuscript (Page 13, Lines 457-460), we have added some descriptions on how antibiotic abuse has led to A. baumannii’s drug resistance. We stated it as below:” In the absence of any effective intervention for A. baumannii infection, a large number of antibiotics have been applied in clinical use. Excessive and combined use of antibiotics can accelerate bacterial resistance. The abuse of antibiotics leads to the increasing mutation of A. baumannii”, pointing out that the absence of effective intervention for A. baumannii infection is one of the reasons for the abuse of antibiotics. Excessive and combined use of antibiotics can accelerate bacterial resistance through the mutation of the bacteria. In this dilemma, the development and application of vaccines have become particularly important.

  • Please expand on A. baumannii’s inclusion in the WHO’s list of critical pathogens to underscore the global health implications.

R:We appreciate the reviewer’s suggestion. In the revised manuscript (Pages 1-2, Lines 35-45), we have expanded on A. baumannii’s inclusion in the WHO’s list of critical pathogens. We emphasized the increasing antibiotic resistance and health-care burden of A. baumannii to highlight its global health implications. Furthermore, we mentioned the estimated proportion of global nosocomial infections caused by A. baumannii.

  • Please highlight the urgent need for vaccines as an alternative to traditional therapeutic drugs, given the pathogen’s resistance.

R: Thank you for your advice. We highlighted the urgent need for vaccines in the prevention and control of A. baumannii in the absence of effective intervention strategies. In the revised manuscript (Page 2, Lines 46-57), we have cited regional data on A. baumannii infections to highlight the enormous harm this bacterium can cause individuals and society. We stated the urgent need for vaccines in the revised manuscript as below: “Vaccines are one of the most robust preventative routes for bacterial infections; therefore, vaccines against A. baumannii are considered as an alternative to traditional therapeutic drugs, because they can significantly reduce the interference of drug resistance.” (Page 2, Lines:62-65)

  • Please explain why subunit vaccines, as opposed to whole-cell vaccines, offer advantages in purity, stability, and efficiency against baumannii.

R: Thank you very much for your suggestion. We added the definition and advantages of subunit vaccine in the revised manuscript (Page 2, lines 66- 76). As we know, subunit vaccines are prepared by prokaryotic expression or chemical synthesis, so their purity is higher. The vaccine candidates are antigens with immunogenicity, so it is more efficient. Finally, due to the removal of ineffective and toxic components during the preparation process, subunit vaccines are safer.

  • Emphasize the potential of OMVs as vaccine candidates, comparing them to existing vaccines like those for meningococcal disease.

R: We appreciate the reviewer’s suggestion. Although OMV-based vaccines in A. baumannii have not been applied in the clinic, an omv-based vaccine against Neisseria meningitis was first licensed for use in 1987. After that, OMVs have been successfully employed as vaccines to prevent meningococcal group B outbreaks. The potent efficiency of these vaccines has been observed in different areas (lines:658-665). The success of OMV vaccines against meningococcal disease indicates the potential of OMVs as vaccine candidates.

  • Please identify any gaps in current research that this paper aims to address, especially in OMV vaccine development.

R: We understand the reviewer’s comment. In the revised manuscript (Page 2, Lines 84-93), we pointed out that there is currently no detailed summarization of the potential application of A. baumannii OMVs in subunit vaccines. This review aims to fill such a gap, and we hope that the generalization of OMVs can provide new ideas for the development of other OMV vaccines.

We stated this comment as below: ”To the present time, there has been no comprehensive generalization of A. baumannii OMVs in subunit vaccine applications. In this review, we summarized the preparation method, pathogenic mechanism, immunogenicity, and immune protective effect of OMVs from A. baumannii. On this basis, we realized the potential application of OMVs in A. baumanniivaccines while also pointing out their shortcomings. In addition to the application of A. baumannii vaccines, the auxiliary role of OMVs in other vaccines is also worth studying. This review provides more comprehensive research directions for OMV vaccine development, including preparation techniques and the efficacy enhancement of other vaccines.”

  • Provide a clearer, more detailed explanation of the biogenesis process of OMVs, possibly with diagrammatic representations.

R: Thank you very much for your precious suggestion. We have elaborated on the biogenesis of OMVs and finished a schematic diagram (Figure 1) to help readers understand. In the revised manuscript, this figure and figure legend were added at the part of “2. Preparation of A. baumannii OMVs” (Page 4).

The figure legend was as follows:” Schematic diagram of biogenesis of A. baumannii OMVs. There are three basic models for the mechanism of biogenesis of A. baumannii OMVs. (A) LPP-PG covalently connects the bacterial OM and the peptidoglycan layer. When LPP is hydrolyzed, reduced, or even absent, the growth rate of the OM will exceed that of the peptidoglycan layer, resulting in OMVs. (B) The second model is based on the occurrence of curvature of the OM, which is associated with the accumulation of phospholipids and peptidoglycans in the OM. (C) LPS is an important biological molecule on the OM of bacteria and is closely related to vesiculation. The refusion of anionic LPS and mutations in LPS-related genes affect the biogenesis of OMVs

  • Please, discuss the mechanisms of OMV formation and cargo selection in more detail, noting any areas of uncertainty.

R: We understand the reviewer’s recommendations. We have discussed the mechanisms of OMV formation and cargo selection in more detail in the revised manuscript. There are three basic models of OMV formation: the lipoproteins (LPP) connection, the accumulation of related components on the outer membrane, and the electric charge of LPS. We added more details to certain mechanism. The first model is due to the weakening or disappearance of LPP connections (line:106-110), the second is the accumulation of substances such as peptidoglycan resulting in changes in membrane curvature, and the last is negatively charged LPS-mediated vesiculation. (line:124-131). Besides, other components also participate in OMV biogenesis(line:143-145). The cargo selection mechanism of OMVs is not very clear, but we know that it is associated with the surrounding environment and the function of OMVs(line:152-159). In addition, the cargo of DNA remains unclear. The procedures of OMV biogenesis and cargo overlap (line:160-174).

  • Please, describe how environmental conditions affect OMV production, linking this to potential vaccine development strategies.

R: Thanks for the reviewer’s suggestion. We have mentioned that the surrounding environment can affect the formation of OMVs. We have offered more depth in how environmental conditions affect OMV production.

In the revised version, the statements are as following (Page 4, Lines 148-159): OMVs play a crucial role in the survival of bacterial cells under abnormal and extreme environmental conditions [20]. Therefore, changes in the surroundings will have an apparent effect on the yield of OMVs. For example, high temperatures can enhance the fluidity of the membrane, speeding up the production of OMVs [21, 22]. The pH fluctuation also regulates membrane integrity and can indirectly affect OMV production via modification of LPS [18]. For example, PmrAB is a system existing in Citrobacter rodentium and is regulated by low pH. Once PmrA is activated, it stimulates the differential ex-pression of pmrC and cptA, both of which catalyze LPS modifications [23]. Based on its susceptibility to the surroundings, the expression of OMVs can be modified artificially to increase the expression level of OMVs. It is a critical part of the manufacture of OMV vaccines.

  • Please offer more depth on the composition of OMVs, including variations among different Gram-negative bacteria.

R: We appreciate the reviewer's suggestion. In the revised manuscript (Page 6, Lines 246-255), we have offered more depth on the composition of OMVs, including variations among different Gram-negative bacteria.

“Almost all Gram-negative bacteria can form OMVs, but their vesicle components varies. It is demonstrated that the biogenesis and composition of OMVs are linked processes. The function of OMVs depends mainly on their cellular components and biological activity. For example, S. Typhimurium provokes OMV biogenesis to shed unfavorable LPS and proteins as an adaptation to new environments, which causes the enrichment of LPS and OM proteins in its OMV [43]. OMVs from commensal Bacteroides are enriched with glycoside hydro-lases in order to degrade environmental polysaccharides. Therefore, the com-position of OMVs can reflect their functions, which can help the development of OMV-based vaccines in different bacteria.”

  • Clarify the role of different components (like phospholipids and LPS) in OMV functionality and their implications for vaccine development.

R: We appreciate the reviewer’s precious advice. Under your guidance, we realized that the previous description of the components of OMVs was too brief. Therefore, in the revised manuscript, we added the introduction of the structure and function of essential components related to vaccine development. The phospholipid layer confines and protects the cargo of OMVs. Besides its function of limitation, the phospholipid bilayer participated in important bacterial activities (line:187-193). Outer membrane proteins play an important role in OMVs. Therefore, we highlighted the description of OmpA, Omp33-36, OmpW, and other proteins. They are essential for the survival and invasion function of the OMVs (line:197-221). We supplemented more details about LPS’s essential role in the biogenetic, virulence, and antibiotic resistance of OMV. We also mention that LPS is an issue that has to solve in vaccine development (line:233-245).

  • Discuss in detail the various methods for preparing OMVs, highlighting the pros and cons of each method.

R: We understand the reviewer’s recommendations. In the revised version, we have described the existing methods of OMV preparation in detail and made a comparison. We pointed out the advantages and disadvantages of each measure, based on which we selected the relatively effective and safe preparation methods (line:277-312). In addition, we also introduce a new method of vesicle preparation (line:313-347), which can undoubtedly provide a new route for the vesicle preparation in A. baumannii

  • Elaborate on how OMVs contribute to A. baumannii’s virulence and pathogenesis, including their role in antibiotic resistance.

R: Thank you very much for your suggestion. We elaborated on the virulence factors of OMVs contributing to A. baumannii’s virulence, which can produce an immune response at the same time (line:356-363). In addition, we added a more detailed mechanism of how OMVs enhance A. baumannii’s virulence (line:370-371). Regarding pathogenesis, we added OMVs’ function of biofilm, which is vital for the survival of the bacteria (line:409-414,453-455). OMVs were involved in multiple pathways as an accelerant of resistance (line:520-522).

  • Provide more insights into the specific mechanisms of drug resistance in A. baumannii and how OMVs play a part.

R: Thank you for your advice. Antibiotic resistance is the main problem in clinical treatment. We have discussed in detail the mechanisms of OMVs in enhancing antibiotic resistance, which includes enzymatic inactivation of antibiotics, target mutation, and reduction of OM transportation (line:473-520). Furthermore, we found that the biofilm function of OMVs also plays a vital role in antibiotic resistance (line:520-522).

  • Discuss the immunogenicity of OMVs in more depth, explaining how they stimulate immune responses.

R: Thanks for the reviewer’s suggestion. Based on the original article, we added the specific process of OMV PAMPs initiating the induction of immune response (line:556-560). We analyzed the reason for the potent immunogenicity of the outer membrane vesicles: the highly conserved composition (line:549-553). In addition, we have increased the body’s response to a secondary bacterial attack (line:572-576).

  • Elaborate on the potential applications of OMVs in vaccine development, including their use as antigen-delivery platforms.

R: We appreciate the reviewer’s suggestion. The potential applications of OMVs in vaccine development can be divided into antigen itself and antigen-delivery platforms. We discussed the function of adjuvant in depth (line:672-682). OMVs can promote enhanced immunogenicity and protect the antigen carried.

“OMV vaccines can also be decorated with specific proteins or polysaccharides from heterologous pathogens. This has been demonstrated through incorporating heterologously expressed outer membrane and periplasmic proteins into OMVs produced by the laboratory and pathogenic E. coli strains [144]. In addition to their simple loading function, engineered OMVs effectively improve the body's immune response. Immunization with E.coli OMVs expressing A. baumannii Omp22 stimulated higher levels of Omp22-specific antibodies than immunization with a recombinant Omp22 protein formulated with alum [14]. OMV decorated with heterologous antigens has been shown to promote enhanced immunogenicity compared to traditional formulations.”

  • Address potential safety and efficacy concerns with OMV-based vaccines, especially regarding LPS content.

R: Thank you for your recommendation. In the revised manuscript (Page 19, Lines 714-737), we have addressed potential safety and efficacy concerns with OMV-based vaccines. We focused on the current issues facing OMV-based vaccines, including their safety and efficacy. The safety of vaccines is mainly related to the LPS component, and there are several ways to address this issue, including detergents and genetic modification. These methods also have room for improvement. As for vaccine efficacy, the main influencing factors are the instability of vesicle composition between strains and the low productivity of native OMVs.

  • Please conclude with a discussion on the broader implications of this research for vaccine development against A. baumannii and other similar pathogens, as well as future research directions.

R: Thank you very much for your suggestion. We have summarized the possible applications of the OMVs in A.baumannii and other pathogens. Based on the adjuvant properties of the OMVs, we supplemented their potential applications in tumor vaccines (line:759-769,776-780). In addition, we proposed some future research directions, including the choice of administration routes in OMVs vaccines (line:785-795).

  • The language and structure used in this section generally require more editing. Finally, addressing these concerns will help to improve the paper.

R: We appreciate the editors’ suggestion. The revised manuscript has been checked and edited by a professional editing company (MDPI). This paper was edited for grammar, phrasing, syntax and punctuation. In addition, many edits were made to further improve the flow and readability of the text.

Reviewer 2 Report

Comments and Suggestions for Authors

vaccines-2749254-peer-review-v1

The manuscript entitled “Outer membrane vesicles from Acinetobacter baumannii: Biogenesis, Functions, and Vaccine Application” by Weng et al., is an interesting document reporting the components, biogenesis, and function of outer membrane vesicles. These outer membrane vesicles in Acinetobacter baumannii may contribute to their potential as a vaccine candidate and for the preparation methods and future directions for their therapeutic developments. The manuscript is compiled nicely and represent a worth reading piece about drug resistant bacterial infections.

Before it goes for publication, I have the following observations to potentially improve the quality of this manuscript:

To my understanding, introduction is quite brief and at least a paragraph should be added e.g., on the control measures and strategies to prevent the A. and baumannii diseases, particularly focusing on the failures of already in practice measures.

The outer membrane proteins are the functional outcomes of the bacterial protein secretion systems; would it not be nice to write a comprehensive synthesis of this aspect. This way, the reader would be connected in bacterial physiology that he/she will be reading in later part of the manuscript.

Table 1: please confine the contents precisely to limits the space/length of the table; line-space can be decreased, bullets should be removed from the ‘mechanisms’ column, 1st letter of the words in this column should be capital, the text in double rows should be comprehended to fit in one row etc.

Figure 1, legend: preparation of what? Please elaborate; the titles/legends should be self-explanatory and explicit.

Figure 1, 2: please add the phrase “schematic diagram/depiction of ----”

L15: use plural practices rather than practice.

L15,21,23 and throughout the manuscript: add space between A. and baumannii.

L304, 313, 316, 318, 324, 326, 328 and throughout the manuscript: add space between the text and [ sign for citations.

Comments on the Quality of English Language

N/A

Author Response

Response to reviewer2

Dear reviewer2,

Thank you for your patience on our submission, and we appreciate the comments concerning our manuscript. Those comments are all valuable for revising and improving our paper. We have studied comments carefully and have made correction which we hope meet with approval. All page numbers refer to the revised manuscript file with tracked changes. The corrections are highlighted with red color in the revised manuscript and listed as below:

(1) In lines 35-70, we have added a paragraph to state the control strategies to prevent A. baumannii diseases, particularly the failures of already in-practice measures.

(2) we have added a comprehensive synthesis of outer membrane proteins (lines 194-232) to state their function in the bacterial protein secretion systems.

(3) In Table 1, we have decreased line space and removed bullets from the “mechanisms” column. 1st letter of the words in this column has been capitalized. The text has been modified to be brief.

(4) Figure 2 legend: To better explain the preparation method of the OMVs in A. baumannii, we added a legend stating the basic procedure of the artificial preparation.

(5) the phrase “schematic diagram/depiction of ----” has been added in figure1,2 and 3.

(6) In line 15, “practice” has been changed to lural practices.

(7) Throughout the manuscript, space has been added between “A.” and “baumannii”.

(8) Throughout the manuscript, space has been added between the text and “[“ sign for citations.

Responses to each comment are listed as below:

The manuscript entitled “Outer membrane vesicles from Acinetobacter baumannii: Biogenesis, Functions, and Vaccine Application”by Weng et al., is an interesting document reporting the components, biogenesis, and function of outer membrane vesicles. These outer membrane vesicles in Acinetobacter baumannii may contribute to their potential as a vaccine candidate and for the preparation methods and future directions for their therapeutic developments. The manuscript is compiled nicely and represents a worth reading piece about drug-resistant bacterial infections.

Before it goes for publication, I have the following observations to potentially improve the quality of this manuscript:

  • To my understanding, introduction is quite brief and at least a paragraph should be added e.g., on the control measures and strategies to prevent the A. and baumannii diseases, particularly focusing on the failures of already in practice measures.

R: We appreciate the reviewer’s precious suggestion. In the revised manuscript, in the part of introduction, we have added more details about the clinical dilemma faced by A. baumannii. In addition to highlighting the importance of vaccines against A. baumannii,we cited more data about A. baumannii worldwide (line:35-45). Meanwhile, we also further elaborate on the advantages of subunit vaccines, which are also the benefit of OMV vaccines (line:66-70).

  • The outer membrane proteins are the functional outcomes of the bacterial protein secretion systems; would it not be nice to write a comprehensive synthesis of this aspect. This way, the reader would be connected in bacterial physiology that he/she will be reading in later part of the manuscript. 

R: Thank you for your valuable suggestion. Outer membrane proteins play a structural role in OMVs. Therefore, as you suggested, a detailed introduction to the function of each outer membrane protein is necessary and can help us better understand bacterial physiology in the later part of the manuscript. We elaborated the function of OmpA, Omp33-36, OmpW and other proteins, including improving bacteria survival and invasion function (line:194-243).

  • Table 1: please confine the contents precisely to limits the space/length of the table; line-space can be decreased, bullets should be removed from the ‘mechanisms’ column, 1stletter of the words in this column should be capital, the text in double rows should be comprehended to fit in one row etc.

R: Thank you for pointing out the error. We have decreased line-space and removed bullets from the ‘mechanisms’ column 1st letter of the words in this column has been capitalized. The text has been modified to be briefer.

  • Figure 1, legend: preparation of what? Please elaborate; the titles/legends should be self-explanatory and explicit.

R: We appreciate the reviewer’s advice. To elaborate on the preparation of OMVs, we have added a legend for Figure 2: “Schematic diagram of artificial preparation of A. baumannii OMVs. Spontaneously released OMVs are isolated from supernatants by ultracentrifugation and filtration. Native AbOMVs are obtained by cutting living bacteria with a high-speed disperser. Detergent OMVs are obtained through detergent. OMVs-prepared high-pressure homogenizers are called SuOMVs. OMVs can also be obtained by ultrasonic bacterial culture and genetic engineering. (sOMVs, spontaneously released OMVs; nOMVs, native OMVs; dOMVs, detergent OMVs; SuOMVs, sucrose extracted OMVs).”

  • Figure 1, 2: please add the phrase “schematic diagram/depiction of ----”

R: Thank you for your suggestion. We have added the phrase “schematic diagram/depiction of ----” in Figure1 ,2 and 3.

  • L15: use plural practices rather than practice. 

R: We are sorry for the typo errors. In line 15, “practice” has been changed to plural “practices”.

  • L15,21,23 and throughout the manuscript: add space between A. and baumannii.

R: Thank you for your suggestion. We have added space between A. and baumannii throughout the revised manuscript.

  • L304, 313, 316, 318, 324, 326, 328 and throughout the manuscript: add space between the text and [ sign for citations

R: Thank you for pointing out the error. We have added space between the text and [ sign for citations throughout the manuscript.

Reviewer 3 Report

Comments and Suggestions for Authors

In the present manuscript, authors review the literature concerning outer membrane vesicles. They describe their formation, their components, their biological role and their potential use as vaccine antigens. In my opinion, the manuscript is well written and covers all aspects. 

I would recommend an addition of a paragraph at the end of the 4th part regarding the patients that could benefit of such a vaccine as far as A. baumannii is an opportunistic pathogen.

I also have some minor recommendations:

line 165: BBVs explain the abbreviation

line 165: "Klebsiella spp." instead of "Klebsiella"

line 304: "factors such as AmpC and OXAs, which..." instead of factor (AmpC)

line 310: NDM-1 instead of NNDM-1

line 314, 322, 324: "Acinetobacter" italics

line 324: Acinetobacter spp.(add spp.)

line 334 "A. baumannii" instead of "Acinetobacter baumannii"

line 337 please explain "exotic" in "exotic antibiotics"

line 341: delete "ability of the"

line 342: "A. baumannii" italics

line 348: "health-care costs" instead of "property"

line 348-349: add "increasing number of vulnerable patients"

line 360: explain "surroundings"

line 370: "susceptibility" instead of "sensitivity"

line 390: delete "known" (there are also other type of APCs)

line 390: "to" instead of "and then"

line 394: rewrite as "Furthermore, the production of anti-inflammatory cytokines, such as IL-10 and IFN-a, is diminished, as reported".

line 401-2: rewrite as "For example, the interaction of TLR4 and lipid A can induce inflammatory cytokine expression....."

line 413:"as shown below " is there a figure? you mean explained?

line 416-7: rewrite as "when the bacteria invade again, the immune system can react effectively and reduce..."

line 438 Neisseria meningitides in italics

line 492: add "or even pan drug-resistant"

Comments on the Quality of English Language

In my opinion, the use of English language is fine.

Author Response

Response to reviewer3

Dear reviewer3,

Thank you for your patience on our submission, and we appreciate the comments concerning our manuscript. Those comments are all valuable for revising and improving our paper. We have studied comments carefully and have made correction which we hope meet with approval. All page numbers refer to the revised manuscript file with tracked changes. The corrections are highlighted with red color in the revised manuscript and listed as below:

(1) In lines 738-751, we have added a paragraph at the end of the 4th part.

(2) In line 313, we have explained the abbreviation “BBVs”.

(3) In line 314, we have replaced “Klebsiella” with “Klebsiella spp.”.

(4) In line 470,“factors such as AmpC and OXAs, which...” replaced “factor (AmpC).”

(5) In line 479, “NNDM-1” has been replaced as “NDM-1”.

(6) In line 460,483,523, we have changed “Acinetobacter” into italics.

(7) In line 524, we have added “spp.” after “Acinetobacter”.

(8) In line 533, "A. baumannii" has replaced "Acinetobacter baumannii".

(9)In line 536,“exotic” in “exotic antibiotics” means that antibiotics are obtained outside the body and are alien to the body. To avoid interference, we have deleted “exotic”.

(10) In line 540, we have deleted “ability of the”.

(11) In line 541, “A. baumannii” has changed into italics.

(12) In line 547, we have replaced “property” with “health-care costs”.

(13) In line 548, we have added “increasing number of vulnerable patients”.

(14) “surroundings” refers to the internal environment of our body.

(15) In line 578, “susceptibility” has replaced “sensitivity”.

(16) In line 608, we have deleted “known”.

(17) In line 610, “to” has replaced “and then”.

(18) In lines 614-615, we have rewritten it as “Furthermore, the production of anti-inflammatory cytokines, such as IL-10 and IFN-a, is diminished, as reported”.

(19) In lines 621-622, we have rewritten it as “For example, the interaction of TLR4 and lipid A can induce inflammatory cytokine expression.....”.

(20) “as shown below” means that the immune procedure has been explained in figure 3.

(21) In lines 636-637, we have rewritten it as “when the bacteria invade again, the immune system can react effectively and reduce...”.

(22) In line 659, we have turned “Neisseria meningitides” into italics.

(23) In line 753, we have added “or even pan drug-resistant”.

Responses to each comment are listed as below:

In the present manuscript, authors review the literature concerning outer membrane vesicles. They describe their formation, their components, their biological role and their potential use as vaccine antigens. In my opinion, the manuscript is well written and covers all aspects. 

  • I would recommend an addition of a paragraph at the end of the 4th part regarding the patients that could benefit of such a vaccine as far as  baumanniiis an opportunistic pathogen.

R: Thank you for your valuable suggestion. To highlight the benefit of OMV vaccines in A. baumannii, we have added a paragraph at the end of the 4th part (line:738-751). Meanwhile, we highlighted the great harm of A. baumannii and conclude the advantages of OMVs as vaccine candidates.

I also have some minor recommendations:

  • line 165: BBVs explain the abbreviation

R: Thank you for pointing out the error. We have explained the abbreviation “BBVs” in line 313.

  • line 165: "Klebsiella spp." instead of "Klebsiella"

R: We are sorry for the typo errors. We have replaced “Klebsiella” with “Klebsiella spp.” (line:313)

  • line 304: “factors such as AmpC and OXAs, which...” instead of factor (AmpC)

R: Thank you for pointing out the error. We have replaced “factor (AmpC)” with “factors such as AmpC and OXAs, which...”(line:470)

  • line 310: NDM-1 instead of NNDM-1

R: We are sorry for the typo errors. “NNDM-1” has been replaced as “NDM-1” (line:479)

  • line 314, 322, 324: “Acinetobacter”italics

R: We are sorry for the typo errors. We have changed “Acinetobacter” into italics. (line: 460,483,523)

  • line 324: Acinetobacter spp.(add spp.)

R: Thank you for pointing out the error. We have added “spp.” after “Acinetobacter” (line:524)

  • line 334 "A. baumannii" instead of "Acinetobacter baumannii"

R: Thank you for your suggestion. “A. baumannii” has replaced “Acinetobacter baumannii” (line:533)

  • line 337 please explain “exotic” in “exotic antibiotics”

R: “exotic” in “exotic antibiotics” means that antibiotics are obtained outside the body and are alien to the body. To avoid interference, we have deleted “exotic” (line:536).

  • line 341: delete “ability of the”

R: Thank you for pointing out the error. We have deleted “ability of the”(line:540).

  • line 342: “ baumannii” italics

R: We are sorry for the typo errors. “A. baumannii” has changed into italics (line:541).

  • line 348: “health-care costs” instead of “property”

R: Thank you for your suggestion. We have replaced “property” with “health-care costs” (line:547).

  • line 348-349: add “increasing number of vulnerable patients”

R: Thank you for your valuable suggestion. We have added “increasing number of vulnerable patients” (line:548)..

  • line 360: explain “surroundings”

R: “Surroundings” refers to the internal environment of our body

  • line 370: “susceptibility” instead of “sensitivity”

R: Thank you for pointing out the error. “susceptibility” has replaced “sensitivity”(line:578).

  • line 390: delete “known” (there are also other type of APCs)

R: Thank you for your valuable suggestion. We have deleted “known” (line:608).

  • line 390: “to” instead of “and then”

R: Thank you for pointing out the error. “to” has replaced “and then” (line:610).

  • line 394: rewrite as “Furthermore, the production of anti-inflammatory cytokines, such as IL-10 and IFN-a, is diminished, as reported”.

R: Thank you for your suggestion. We have rewritten as “Furthermore, the production of anti-inflammatory cytokines, such as IL-10 and IFN-a, is diminished, as reported” (line:614-615).

  • line 401-2: rewrite as “For example, the interaction of TLR4 and lipid A can induce inflammatory cytokine expression.....”

R: Thank you for the reviewer’s suggestion. We have rewritten as “For example, the interaction of TLR4 and lipid A can induce inflammatory cytokine expression.....”(line:621-622).

  • line 413: “as shown below “is there a figure? you mean explained?

R: “As shown below” means that the immune procedure has been explained in the figure 3.

  • line 416-7: rewrite as “when the bacteria invade again, the immune system can react effectively and reduce...”

R: Thank you for pointing out the error. We have rewritten as “when the bacteria invade again, the immune system can react effectively and reduce...”(line:636-637).

  • line 438 Neisseria meningitides in italics

R: Thank you for pointing out the error. We have turned “Neisseria meningitides into italics (line:659).

  • line 492: add “or even pan drug-resistant”

R: Thank you for the reviewer’s suggestion. In the revised manuscript, we have added “or even pan drug-resistant” (line:753).

Round 2

Reviewer 1 Report

Comments and Suggestions for Authors

No comments